# Occurrence of Diarrhea and Feeding Practices among Children below Two Years of Age in Southwestern Saudi Arabia

**DOI:** 10.3390/ijerph17030722

**Published:** 2020-01-22

**Authors:** Ayed A. Shati, Shamsun N. Khalil, Khalid A. Asiri, Abdulaziz Ahmed Alshehri, Yazeed A. Deajim, Mohammad S. Al-Amer, Hassan J. Alshehri, Abdulaziz Abdullah Alshehri, Fahad S. Alqahtani

**Affiliations:** 1Department of Child Health, College of Medicine, King Khalid University, Abha 61421, Saudi Arabia; 2Department of Family and Community Medicine, King Khalid University, Abha 61421, Saudi Arabia; shamsun203@gmail.com; 3College of Medicine, King Khalid University, Abha 61421, Saudi Arabia; Fefe20@windowslive.com (K.A.A.); alshehri.abdulaziz17@gmail.com (A.A.A.); Yazeedad@gmail.com (Y.A.D.); Moh1417d@yahoo.com (M.S.A.-A.); Hassanalshehri10@gmail.com (H.J.A.); Abdulazizalshehri.com@gmail.com (A.A.A.); fahad.s.m34@gmail.com (F.S.A.)

**Keywords:** feeding practices, diarrhea, children of two years of age, Aseer region

## Abstract

Growing evidence suggests that feeding practices in early childhood play a major role in the occurrence of childhood diarrhea. However, there is a lack of information regarding feeding practices and its relationship with occurrences of diarrhea in young children from Saudi Arabia. The present study is aimed to measure the prevalence of diarrhea and assess its relationship with feeding practices among children between two months and two years of age in Saudi Arabia. A cross-sectional study was carried out in two large cities in the Aseer region in southwest Saudi Arabia. A total of 302 mothers attending well-baby clinics across six primary health centers were included. A structured questionnaire was used to collect data. Factors associated with diarrheal disease were identified by multivariable logistic regression analysis. The prevalence of diarrhea among children during the study period was 56.3% (95% CI: 50.7%–61.8%). Only 15.9% of children in our study were exclusively breastfed. The occurrence of diarrhea was significantly associated with age 7–12 months (aOR = 2.64, 95% CI: 1.42–4.91). We found that diarrhea was prevalent among children between two months and two years of age, and that exclusive breastfeeding was not a common practice in this region. Health education programs should be directed towards mothers to improve rates of breastfeeding, weaning practices, food hygiene, and childcare. Special attention and support should be provided for working mothers.

## 1. Introduction

Each year, about two million children below 5 years of age die from diarrhea around the world; 80% of these deaths occur in the first two years of life. Although there has been a significant decline in childhood mortality rates from diarrhea, it still poses a major public health concern, especially in developing countries [1,2].

Mortality is the most severe consequence of diarrheal disease, which also has long-term negative impacts in the first two years of life [3]. The long-term impacts include growth failure, impaired physical fitness, decreased cognitive ability, and poor performance in school [4,5]. Diarrheal diseases result from the additive effects of socioeconomic, environmental, and behavioral factors, and improper child feeding practices. Feeding practices like early breastfeeding initiation, exclusive breastfeeding, complementary feeding initiation, complementary food hygiene, hand washing at the time of feeding, and child vaccination are factors associated with childhood diarrhea [3,6,7]. An increasing number of studies has confirmed that breastfeeding acts as a protective factor against diarrhea, which reduces the occurrence of childhood diarrhea and its severity [8,9,10].

The main factors responsible for diarrheal disease across the world are inadequate drinking water, sanitation and hygiene practices, and a tropical climate [11,12]. In Saudi Arabia, these factors are absent due to the hot and dry climate, which is unfavorable for diarrheal diseases [13]. The population also has access to clean drinking water and sanitary latrines. Nonetheless, the reported prevalence rate of diarrhea in children up to two years of age in Saudi Arabia is 25%, i.e., similar to the global average [14].

Previous studies conducted on diarrheal diseases in children in Saudi Arabia reported that children from the southern region have higher rates of diarrhea compared to those in other parts of the county [14,15]. However, both studies provided limited information regarding the causes. Thus, we aimed to estimate the prevalence of diarrhea and its correlation with feeding practices. This study will also allow us to bridge the knowledge gap by generating epidemiological information by which to guide proper formulation of prevention and control programs in this area.

## 2. Methods

### 2.1. Study Design and Settings

This cross-sectional study was carried out between June and July 2018 in Abha and Khamis Mushait City in Aseer Province in south-west Saudi Arabia. Abha is the capital city of Aseer Province, and is located in the southern region of Saudi Arabia at an elevation of about 2270 m above sea level. The climate of Abha is semi-arid, and is affected by city’s high elevation. Khamis Mushait is a large city located east of Abha with a hot desert climate. The populations of Abha and Khamis Mushait are 210,886 and 387,553, respectively [16].

### 2.2. Subjects

Using a simple random sampling method, a total of six large primary health care centers (PHCC), three from Abha and three from Khamis Mushait, were selected for the study.

The sample size was calculated by using Epi info software version 7.2 (CDC, Atlanta, GA, USA) with the expected prevalence of diarrhea of 25% [14] at a 95% confidence interval (CI), 80% power, and acceptable error 5%. The estimated sample size was 285 children. A consecutive sampling technique was used to select the subjects, and an equal number of subjects were considered from each PHCC. Data were collected from 302 mothers having children up to two years of age who attended the well-baby clinic in selected PHCCs. In cases where the family had more than one child between two months and two years of age, the youngest child was included in the study.

### 2.3. Data Collection Tool

A structured questionnaire was used to collect the data. The questionnaire included socio-demographic criteria, feeding practices of the children, and the occurrence of diarrhea. In this study, diarrhea is defined as three or more loose or liquid stools passed in a 24-h period [2]. Diarrhea was measured based on mothers’ responses (yes or no) to whether the child of 2–24 months of age had suffered diarrhea in the one-year period before the date of data collection. The feeding practices of children included history of breastfeeding and artificial feeding status. The breastfeeding status of children was classified as early initiation of breastfeeding (was breastfed immediately after birth and received colostrum at birth), exclusive breastfeeding (was exclusively breastfed for 6 months and no prelacteal feeding at birth), and continuation of breastfeeding (present status and duration of breastfeeding). Regarding artificial feeding, variables included types of formula feeding, home-made food, feeding frequency, reason for formula feeding at birth, and clean hands before making a meal.

To test the comprehensiveness and language of the questionnaire, it was pretested with 20 mothers who had similar characteristics out of the selected PHCCs. The data of the pretest were omitted from the main analysis. Based on the feedback of the participants, modifications were made to the questionnaire.

### 2.4. Data Collection Method & Ethical Consideration

Data were collected through face-to-face interviews with the mothers, as conducted by the researchers. The data collectors were provided with one day’s training on data collection methods. The principal researcher was responsible for the overall coordination of the whole data collection process and checking of the data quality during the data collection period. Preceding the interview, informed consent was obtained from the mothers after they were explained the objective and method of the study. Anonymity and confidentiality were preserved through all the steps of the research. The study protocol was approved by the Research Ethics Committee (REC) of King Khalid University in Abha, Kingdom of Saudi Arabia (REC# 13 April 2018).

### 2.5. Statistical Analysis

Data were analyzed using SPSS version 22.0 (IBM, North Castle, NY, USA). Descriptive statistics were presented as frequencies and percentages. A bivariate analysis was conducted using chi-square test. Logistic regression analysis was used to determine the variables which could predict the occurrence of diarrhea in a multivariable context. The independent variables that were found to be significant in the bivariate analysis were subjected to regression model, and a backward stepwise method was used for the multivariate analysis. The results obtained from multivariate analysis are presented as adjusted odds ratio (aOR) with concomitant 95% CI. A *p*-value of less than or equal 0.05 was considered statistically significant.

## 3. Results

### 3.1. Characteristics of the Study Children

The study comprised of a total of 302 children, of which 52% (157) were female. The age of the children ranged between 2 to 24 months; 215 (71.2%) were aged between 2–11 months. The mean age of the children was 10.4 months, with a standard deviation of 6.4. The majority of the children (*n* = 257, 85.1%) came from families where there was only one child below two years of age, and 16 (5.3%) children came from families with three children up to two years of age. Less than a third (27.8%) of the children were the first child in the family. Most of the children (92.1%) had completed vaccinations. The mean age of the mothers was 30.3 ± 5.5 years, and nearly 55% of the mothers were aged 25–34 years. The age of the mothers ranged between 18 to 45 years. More than half of the mothers had received above secondary-level education. However, only 31.1% of the mothers were employed at the time of data collection. Less than half (48.3%) of the households had monthly incomes of more than ten thousand Saudi Riyal (Table 1).

### 3.2. Feeding Practices

At the time of survey, around 7% of the children were fed breast milk, while 30.8% were fed formula; 63.2% received a combination of breast milk, formula, and home-cooked food. Regarding feeding practices at birth, more than half (51.7%) of the children received breast milk within one to two hours immediately after birth. Only 48 (15.9%) were exclusively breastfed for the first six months, while 30.8% were not breastfed.

The most common reason reported by the mothers for giving formula feeding at birth was insufficient breast milk (84.2%). The majority of mothers washed their hands before they made food for their child (Table 2).

### 3.3. Occurrence of Diarrhea in the Study Children

Table 3 presents the occurrence of diarrhea among two-year old children. The prevalence of diarrhea was 56.3% in the preceding year (95% CI, 50.7, 61.8%). The age-specific prevalence of diarrhea was the highest, at 40% among children aged 7–12 months (95% CI:32.9–47.5%). Mean episode of diarrhea was 2.1 ± 1.8. Over half (57.1%) of the children had one episode of diarrhea, while 30.0% children experienced 2–3 episodes. The majority of the children who had diarrhea received treatment either from outdoor clinics of primary health care centers or at home. Only 9.4% of children were hospitalized, and the mean duration of stay in hospital was 1.5 ± 1.1 days.

### 3.4. Factors Associated with Childhood Diarrheal Disease

The factors significantly associated with diarrhea in the bivariate analysis are shown in Table 4. In terms of feeding practices, only exclusively breastfeeding seemed to have a protective effect against diarrhea (OR = 0.447, 95% CI:0.261–0.911). No other feeding practices had any significant association. Other factors like socio-demographic factors such as the age of the children, order of the child in the family, and the occupation of the mother were significantly associated with childhood diarrhea. Rates of diarrhea were significantly higher among older children (*p* = 0.001) and amongst children who were the first born in the family (*p* = 0.04). Children whose mothers were employed showed significantly higher rates of diarrhea compared with their counterparts (*p* = 0.001).

A multivariate analysis was performed to identify the risk factors that are independently associated with occurrence of diarrhea (Table 5). When controlling for other variables, the order of the child in the family and breastfeeding status appeared to be insignificant. However, the age of the children and mother’s occupation were found to be significant independent predictors of childhood diarrhea. Children aged 7–12 months were almost three times more likely to develop diarrhea compared with the children aged 2–6 months (aOR = 2.64, 95% CI: 1.42–4.91). The risk of diarrhea was two times higher among children who had working mothers (aOR = 2.496, 95% CI, 1.43, 4.24) compared with children whose mothers were not employed.

## 4. Discussion

This study examined the contributing factors of diarrhea in children between two months and two years of age, with special reference to feeding practices and socio-demographic factors. The prevalence of diarrhea among the study population was 56.3% in the year preceding the data collection. Compared to previous studies, this prevalence is substantially higher. This can be attributed to the fact that our study collated data for one year, whereas past studies have reported prevalence rates for only two weeks. Our study reported a mean of two per child per year for diarrheal episodes, which is similar to rates reported in previous studies conducted in Saudi Arabia [14,15].

Breastfeeding is universally endorsed as the best way of feeding infants [17,18,19,20]. However, exclusive breastfeeding is not a common practice among Saudi mothers, and is usually combined with infant formula food. Only 15.9% of the children in this study were exclusively breastfed, which is lower than WHO recommendations [17]. In our study, exclusive breastfeeding was found to be a protective factor against diarrhea, which is in agreement with other studies around the world [21,22]. The protective effect of exclusive breastfeeding against infectious disease-related morbidity in infancy in Bangladesh has been reported [21]. A Nigerian study also noted that predominantly- and partially-breastfed infants were more likely to have reported diarrhea than exclusively-breastfed infants [22]. Many infants who cannot be breastfed for various reasons are given infant formulas [23,24]. In this study, artificial feeding, particularly formula feeding, was found to be a common practice among children below six months of age. Previous studies have reported that the early introduction of complimentary food and formula feeding increases the risk of food-borne infections [25,26,27,28,29,30,31,32,33]; however, we did not find any such association. In addition, studies have reported that the higher rates of episodes of child diarrhea are related to poor sanitary conditions and hand washing practices [34]. We found that most of the mothers practiced hand washing, and that it did not have an impact on the occurrence of diarrhea. The other aspects of food preparation like cleaning of feeding bottles/utensils were not considered, and need further research.

Rates of diarrhea were significantly higher among children aged 7–12 months and among the first born in the family, which is in agreement with previous studies. Episodes of diarrhea were more common in children older than six months of age [14,35]. It is a common phenomenon that children above six months of age start physical movements and explore things around them. In many cases, they put things in their mouth, exposing themselves to various infections. Being the first child in the family, maternal inexperience in childcare results in higher rates of diarrheal infections. The mother might lack knowledge of the risk factors of diarrhea and its route of transmission. Although some studies have found low maternal educational levels to be significantly associated with childhood diarrhea [35], we did not find such an association in our study.

Variable effects of vaccination on occurrences of diarrhea have been reported in previous literature. For example, a study in Bangladesh reported no significant impact of rotavirus vaccination on incidence of rotavirus-induced diarrhea [36], while another study in Mozambique reported a decrease in rotavirus positivity and diarrhea hospitalizations after the introduction of the vaccine [37]. In this study, most of the children had received rotavirus vaccination as part of the expanded program of immunization in Saudi Arabia. We found no correlation between vaccination and diarrhea. The cross-talk of immunological mechanisms of diarrheal disease and vaccination was beyond the scope of this study. However, the majority of the children who had diarrhea received treatment either from outdoor clinics of primary health care centers or at home; one in ten children were hospitalized for treatment of diarrhea. This is lower than the rate observed in a study conducted in Bangladesh, where one out of every six cases of childhood diarrhea required hospitalization [24].

## 5. Strengths and Limitations

This is the first study in the Aseer region to establish the relationship between the occurrence of diarrhea and feeding practices among children of two years of age. One of the limitations of the study is that the results may not be generalized for a large country like Saudi Arabia. Some methodological limitations of this study should also be considered. The study could not establish a clear temporal association between feeding practices and diarrhea and seasonal differences in the occurrences of diarrhea, as only cross-sectional data were used. Secondly, recall or measurement bias may have occurred, as the exposure and outcome variables were based on self-reporting. The study may have been affected by social desirability bias, since cases of diarrhea were considered based on the reports from mothers, without confirmation from physicians. The analysis may have overestimated or underestimated the prevalence of diarrhea and its association between child feeding practices. Thirdly, unmeasured confounding factors such as preterm, small for gestation, the impact of environment, culture, family conditions, food preparation, household hygiene, or cleanliness and sanitation may have affected the results.

## 6. Conclusions

Despite the absence of the typical environmental factors conducive for diarrhea in Saudi Arabia, the rates are comparable to those of any other developing country. This can be attributed to other factors, particularly maternal factors like the working status of the mother and the early introduction of complementary feeding without proper hygiene. Despite this, exclusive breastfeeding is still not a common practice. These findings point to the need for better health education advice for mothers regarding exclusive breastfeeding, weaning practices, food hygiene, and child care.

## Figures and Tables

**Table 1 ijerph-17-00722-t001:** Background characteristics of the children.

Characteristics	Number	%
**Gender**		
Boy	145	48.0
Girl	157	52.0
**Age of Child**		
2–6 months	106	35.1
7–12 months	109	36.1
>12 months	87	28.8
**Children up to Two Years of Age in the Family**		
One child	257	85.1
Two children	29	9.6
Three children	16	5.3
**Order of the Child in the Family**		
First child	84	27.8
Not the first child	218	72.2
**Age of the Mothers**		
18–24 years	50	16.6
25–34 years	165	54.6
≥35 years	87	28.8
**Mothers’ Education Level**		
Below secondary	33	10.9
Secondary	98	32.5
Above secondary	171	56.6
**Maternal Employment Status**		
Employed	94	31.1
Un-employed	208	68.9
**Fathers’ Education**		
Below secondary	29	9.6
Secondary	98	32.5
Above secondary	175	57.9
**Monthly Family Income**		
<5000 SR	46	15.2
5000–10,000 SR	110	36.4
>10,000 SR	146	48.3
**Vaccination Status**		
Vaccination complete	278	92.1
Vaccination not complete	24	7.9

**Table 2 ijerph-17-00722-t002:** Feeding Practices of the children.

Feeding Practices	Number	%
**Type of Feeding at the Time of Survey**		
Breast feeding only	18	6.0
Formula feeding only	93	30.8
Breast milk, formula & home cooked food	191	63.2
**Start Breast Feeding at Birth**		
On the day of birth	158	52.3
2 to 3 days after birth	51	16.9
Not breastfed	93	30.8
**Duration of Breast Feeding**		
≤6 months	156	51.7
7–12 months	40	13.2
>12 months	13	4.3
No breast feeding	93	30.8
**Exclusive Breast Feeding**		
Exclusively breastfed up to 6 months	48	15.9
Not exclusively breastfed	161	53.3
Not being breastfed	93	30.8
**Reason for Formula Feeding at Birth (*n* = 254)**		
Insufficient breast milk	214	84.2
Mother was ill	20	7.9
Child was ill	20	7.9
**Clean Hands with Soap and Water before Making a Meal**		
Yes	263	87.1
No	39	12.9

**Table 3 ijerph-17-00722-t003:** Occurrence of diarrhea in the children.

Occurrence of Diarrhea	Number	%
**Prevalence of Diarrhea**		
Yes	170	56.3
No	132	43.7
**Age Specific Prevalence of Diarrhea (*n*-170)**		
2–6 months	44	25.9
7–12 months	68	40.0
13–24 months	58	34.1
**Episode of Diarrhea (*n*-170)**		
One episode	97	57.1
2–3 episodes	51	30.0
>3 episodes	22	12.9
**Admitted in Hospital (*n*-170)**		
Yes	16	9.4
No	154	90.6

**Table 4 ijerph-17-00722-t004:** Factors associated with childhood diarrheal disease.

Risk Factors	Diarrhea		*p*-Value
	Yes *n* (%)	No *n* (%)	
**Age of Child**			
2–6 months	44 (41.5)	62 (58.5)	
7–12 months	68 (62.4)	41 (37.6)	0.001
13–24 months	58 (66.7)	29 (33.3)	
**Order of the Child in the Family**			
First child	55 (65.5)	29 (34.5)	0.046
Not the first child	115 (52.8)	103 (47.2)	
**Maternal Employment Status**			
Working	68 (72.3)	26 (27.7)	0.001
Non-working	102 (49.0)	106 (51.0)	
**Breast Feeding Status**			
Exclusive breast feeding	20 (41.7)	28 (58.3)	0.023
Not exclusive breast feeding	151 (59.4)	103 (40.6)	

**Table 5 ijerph-17-00722-t005:** Predictor of diarrhea among the children.

**Variables**	**aOR (95% CI)**	***p*-Value**
**Age of the Children**		
2–6 months	*Ref*	
7–12 months	2.64 (1.42–4.91)	0.004
13–24 months	1.08 (0.589–1.98)	0.819
**Mothers’ Occupation**		
Not working	*Ref*	
Working	2.46 (1.43–4.24)	0.001

Ref = reference group, aOR = adjusted Odds, Ratio, 95% CI = 95% confidence interval.

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
