# Peer review of "Occurrence of Diarrhea and Feeding Practices among Children below Two Years of Age in Southwestern Saudi Arabia"

_ijerph, 2020, doi:10.3390/ijerph17030722_

Round 1

Reviewer 1 Report

This revised manuscript appears significantly improved and would recommend acceptance.

Reviewer 2 Report

I think you should remove the number of feeds as this does not add anything to the article and is difficult to define/measure. This is highly dependent on the age of the child; young children tend to feed more frequently. Also how is a feed defined; for example if I give breastmilk and then food  (or vice versa) is this one feed or two. Defining a breastfeed is notoriously difficult, some infants nurse for long periods and sleep at the breast so it can be difficult to define. Also if infants cluster feed or co-sleep with night feeding this is difficult to measure.

Line 24 – remove the word “more” from before “prevalent”

Line 175   “breastfeeding is not a common practice and is usually combined with infant formula” – do you mean “exclusive breastfeeding is not a common practice”

Reviewer 3 Report

Dr Shati et al present data on the association of infant feeding practices and incidence of diarrheal illness.

The authors have improved the manuscript according to previous suggestions.

The study design is appropriate. The questionnaire evaluation is sound. 

The authors present the data in an appropriate way,  discuss the applicability to the specific region and also limitations and possible confounding factors in data collection and ascertainment of the associations found.

Author Response

This manuscript is a resubmission of an earlier submission. The following is a list of the peer review reports and author responses from that submission.

Round 1

Reviewer 1 Report

 Since there is paucity of literature regarding feeding practices and diarrhea among young children in Saudi Arabia, this work will be helpful. I have few minor suggestions.

Line 244 - Infants less than 6 months also generally have less infections from maternal antibodies and also from breastfeeding (may not be applicable to this study population given the lower prevalence) Did the authors consider the infant characteristics such as preterm, small for gestational age, formula characteristics (whole milk protein, soy or hypoallergenic formulas?).  Typographical errors are noted such as line 270 (c of), line 232 (Indonesia is the correct word) etc.. 

Reviewer 2 Report

This manuscript describes a study of feeding practices In children aged 2 months – 2 years and the association with diarrhea in Saudi Arabia.

I am not convinced about separating the children by 0-12 month and 12-24 months in the analysis. Diarrhoea is usually low in exclusively breastfed infants. On introducing either formula or solids is when diarrhoea is more likely to occur.  I think this is what you should have been looking at rather than just splitting by age of children. Yes older children will have more diarrhea, this is expected.

Also you have such a large age spread 2-24 months – there is also a huge difference not only in food but also the environment. A two month baby lies still whereas older children can crawl/walk around and put things in their mouths – no matter how much hygiene we have as parents.

The term artificial feeding is used incorrectly  - this applies to formula only. Other foods should be referred to as complementary feeding.

Introduction

Line 57 – what do you mean by the rate of diarrhea is 25% in Saudi Arabia, does this mean 25% of children have it within a year? What was the rate in the other Saudi Aribain study?

Line 60 – what do you mean by “more than a decade”. Where these studies carried out over a decade?

Methods

Line 98 – did you mean washing hands before preparing infant formula? Did you also ask women about washing hands before preparing other food?

Results

You do not need to include everything from your tables in the text, only state what is noteworthy

3.2 feeding practices – this section is confusing you state 63% of children received breast milk and artificial foods. What are artificial foods – do you mean formula milk or complementary foods. You also need to make it clear which infants have milk only (breastmilk and/or formula) and which are receiving complementary feeding.

You state amongst women who are artificially feeding formula -  ‘only’ is popular (80%), but this contradicts that 63% received breast milk and artificial food (if this is formula milk)

Line 134 What is the relevance of the income groups you split your participants into – is this mean population income?

Line 148 – what is the relevance of 5 formula feeds?

Discussion

Why is your diarrheal rate (50%) so much higher than that seen previously – you state it was similar to previous studies, yet you cited 25% was seen previously, this is not similar to 50%?

Line 200 – here you state in SA most mothers combine BF with infant formula – but this contradicts your own data of where women provide infant formula, 80 % of women provide formula alone.

Line 219 “we found a significant association between complementary feeding and diarrhoea" – this is not in your results.

Reviewer 3 Report

Dr Shati and colleagues present a questionnaire-based study from Saudi Arabia with a view to elicit risk factors for diarrheal disease. Particular emphasis has been put on breastfeeding and other maternal demographic factors. 

Overall the article is appropriately structured. The methodology is adequate and the statistical analysis supports the authors` conclusions. 

Formal aspects:

The introduction should be shortened. 

The methods section explains the questionnaire design and validation, the sample size estimation is adequately explained. A quick explanation of the immunization schedule in the region would be helpful for the reader.

The discussion needs to be shortened significantly. The first paragraph (LL187-192) should be omitted as it repeats the introduction.

The conclusions in the discussion are supported by the results, with acknowledgement of the potential limitations. It could be discussed in greater detail how the benefits of breastfeeding could be conveyed to the mothers of childbearing age in culture-sensitive way.

Overall, quite a significant number of grammatical and spelling changes need to carried out. 

Suggestions:

L14: feeding practices play ...

L16: ..relation to the occurrence of ...

L 38: deaths occur

L225: ...studies in developing countries .... ; and others